# Aligning Silhouette Topology for Self-Adaptive 3D Human Pose Recovery

**Mugalodi Rakesh**[1*]    **Jogendra Nath Kundu**[1*]    **Varun Jampani**[2]    **R. Venkatesh Babu**[1]

[1]Indian Institute of Science, Bangalore    [2]Google Research

## Abstract

Articulation-centric 2D/3D pose supervision forms the core training objective in most existing 3D human pose estimation techniques. Except for synthetic source environments, acquiring such rich supervision for each real target domain at deployment is highly inconvenient. However, we realize that standard foreground silhouette estimation techniques (on static camera feeds) remain unaffected by domain-shifts. Motivated by this, we propose a novel target adaptation framework that relies only on silhouette supervision to adapt a source-trained model-based regressor. However, in the absence of any auxiliary cue (multi-view, depth, or 2D pose), an isolated silhouette loss fails to provide a reliable pose-specific gradient and requires to be employed in tandem with a topology-centric loss. To this end, we develop a series of convolution-friendly spatial transformations in order to disentangle a topological-skeleton representation from the raw silhouette. Such a design paves the way to devise a Chamfer-inspired spatial topological-alignment loss via distance field computation, while effectively avoiding any gradient hindering spatial-to-pointset mapping. Experimental results demonstrate our superiority against prior-arts in self-adapting a source trained model to diverse unlabeled target domains, such as a) in-the-wild datasets, b) low-resolution image domains, and c) adversarially perturbed image domains (via UAP).

## 1   Introduction

Human pose recovery has garnered a lot of research interest in the vision community. It finds extensive use in a wide range of applications such as, augmented reality, virtual shopping, human-robot interaction, *etc*. Recent advances in human pose recovery is largely attributed to the availability of 3D pose supervision from large-scale datasets [19, 42, 57]. Although, the models achieve superior performance on in-studio benchmarks, they usually suffer from poor cross-dataset performance. Training on synthetic and in-studio datasets, which lack diversity in subject appearance, lighting, background variation, among more, can inherently induce a domain-bias [36, 27] thereby restricting generalizability. This calls for the question: how to bridge performance gaps across diverse data domains?

Some works use weaker forms of supervision such as 2D pose keypoints, either directly [23, 43] from the manually annotated large-scale datasets [1, 21] or indirectly [5, 26] via an off-the-shelf 2D pose detector network. Further, multi-view [51, 20] cues have also been used to learn a reliable 2D-to-3D mapping. However, acquiring such additional supervision often comes at a cost, such as laborious manual skeletal-joint annotation or cumbersome calibrated multi-camera setups. This severely limits the deployability of a model in a novel, in-the-wild target environment. Images in a target deployment scenario can vary from those used in training setup in several ways, from simple changes in lighting or weather, to large domain shifts such as thermal or near-infrared (NIR) imagery. It is highly impractical to gather pose annotations for every novel deployment scenario.

---

[*]Equal contribution. | Webpage: https://sites.google.com/view/align-topo-human

35th Conference on Neural Information Processing Systems (NeurIPS 2021).

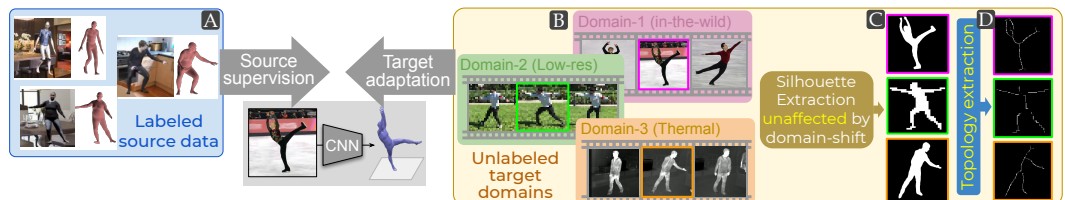

Figure 1: The proposed self-supervised adaptation relies only on silhouette supervision (unaffected by domain-shift) to adapt the source trained pose regressor to novel target deployment scenarios. Topology extraction from raw foreground silhouettes is one of the key design aspects of our framework.

Motivated by these observations, we aim to realize a self-adaptive monocular 3D human pose recovery framework that does not rely on any strong ground-truth (2D/3D pose) or auxiliary target supervision (multi-view, or depth). The proposed adaptation framework (Fig. 1) aims to transfer the task knowledge from a labeled source domain to an unlabeled target (Fig. 1B), while relying only on foreground (FG) silhouettes. We observe that the traditional FG segmentation techniques are mostly unaffected by input domain shifts (Fig. 1C) and thus the FG masks are usually robust and easy to obtain in diverse target scenarios. Consider a deployment scenario where a static camera is capturing a moving object. Here, one can rely on classical vision based background subtraction techniques to obtain a FG silhouette while being considerably robust to a wide range of domain shifts (including gray-scale, thermal, NIR imaging, etc). Certain deep learning based class-agnostic FG segmentation methods [61, 39] (motion and salient-object segmentation methods) also exhibit reasonable robustness to domain-shifts. Hence, an adaptation framework that can effectively use silhouette supervision would open up the scope for adaptation to diverse deployment scenarios.

Prior human mesh recovery works [26, 20, 23, 24] have demonstrated considerable performance gains by training with articulation-centric annotations such as 3D and 2D pose supervision. At the same time, silhouettes have found a relegated use, mostly towards enforcing auxiliary shape-centric objectives [9, 47]. Although these weaker objectives employ silhouettes, they do not offer strong supervisory signal in isolation, hence are not self-sufficient. In this paper, we aim to realize a stronger, self-sufficient and articulation-centric supervision from raw silhouettes. The key challenge is that the articulation information is usually lost in direct pixel-level losses, as raw silhouettes mostly provide 2D shape information. Our key insight is to extract a skeleton-like 2D topology information from silhouettes (see Fig. 1D), which can provide a stronger articulation-centric supervision. We further propose a topological alignment objective that is inspired from 3D Chamfer distance [11] and can directly work on binary images. We leverage this objective to drive our target adaptation training using just silhouettes, and demonstrate the applicability of our framework across various domains. We make the following main contributions:

- We develop a series of convolution-friendly spatial transformations in order to disentangle a topological-skeleton representation from the raw silhouette.
- We devise a Chamfer-inspired spatial topology alignment loss via distance field computation, while effectively avoiding any gradient hindering spatial-to-pointset mapping.
- Our self-supervised adaptation framework achieves state-of-the-art performance on multiple challenging domains, such as low-resolution domain (LR), universal adversary perturbed domain (UAP), in-studio Human3.6M [19], and in-the-wild 3DPW [58].

## 2 Related work

**Human mesh recovery.** Supervised human mesh recovery on monocular RGB images has been well explored in recent years, works such as [26, 23, 24, 47] achieve impressive performance on standard benchmarks. These works heavily rely on large-scale annotated in-studio datasets [19, 42] for the bulk of their training, however they report and agree on the lack of generalized performance on unseen in-the-wild data. Most works [23, 26, 49, 47, 24, 33, 29, 5] try to address the issue by utilizing manually annotated 2D supervision from in-the-wild datasets [22, 1, 35], and some additionally fine-tune their models by enforcing temporal [24, 17] or multiview [34, 49, 17] constraints. Kolotouros *et al*. [26] has also explored integrating pre-trained off-the-shelf 2D pose detectors into the training loop, aimed towards improving in-the-wild performance.

Despite the diversity in approaches, prior works mainly focus on applicability to a single target domain i.e. the ideal in-the-wild RGB data. Due to such a viewpoint, even the weakly-supervised approaches [10, 55, 47] render themselves irrelevant towards extending to new unlabeled target

domains, as their methods implicitly assume access to significant amount of annotated samples from the final target domain. Xu *et al.* [60] proposed to overcome the performance-gap due to low-resolution (LR) imagery, but required labeled targets with simultaneous access to high-resolution source data. Later, Zhang *et al.* [62] proposed to address the domain induced performance degradation via self-supervised inference stage optimization. But their requirement of 2D pose keypoints as input, all the while only addressing 3D pose distribution shifts, severely restricts their applicability. Recognising this deficiency in prior works, our aim is to provide a self-supervised framework which can facilitate the overcoming of performance gaps across a wide array of domains.

**Use of silhouettes.** Silhouettes have found extensive use in works which aim to obtain 3D human body shape from images. Sigal *et al.* [53] is one of the earliest works attempting to estimate the high quality 3D human shape, via fitting a parametric human-body model (*i.e.*, SCAPE [2]) to ground truth silhouettes. Subsequently, Guan *et al.* [15] used silhouettes, shading and edges as cues towards the body-shape fitting process, but required a user-given 2D pose initialization. Lassner *et al.* [33] proposed a silhouette based fitting approach for SMPL [38]

Table 1: Characteristic comparison of our method against prior arts, separated by access to direct (paired) supervision for adaptation to new domains. MV: multi-view, Sil.: silhouette

| Model-based Methods | Paired sup. | | | Self-sup. Adaptation |
|---|---|---|---|---|
| | 2D/3D pose | MV cams | Sil. | |
| [5, 23, 26] | ✓ | ✗ | ✗ | ✗ |
| [34, 49, 17] | ✓ | ✓ | ✗ | ✗ |
| [33, 15, 47, 55] | ✓ | ✗ | ✓ | ✗ |
| Ours | ✗ | ✗ | ✓ | ✓ |

human-body model, by using an off-the-shelf segmentation network. These works primarily aim for shape fitting based optimization and attempt to fit a 3D parametric mesh to a set of 2D evidence. However, solving these iterative optimization formulations for each and every instance can be prohibitively slow at test-time, and are also prone to getting stuck in local minimas. In contrast to these shape-centric works, we primarily aim to recover accurate posture of the person and consider shape estimation as a useful by-product of our formulation.

A few works [55, 47, 30] formulate encoder-decoder based architectures with silhouette and 2D keypoint reconstruction as primary training objectives for their final regressor network. Silhouettes have also been used [56, 29, 10, 54, 3] as weak forms of auxiliary supervision usually in tandem with optical-flow, motion or other self-supervised objectives, some of these even relying on synthetic data for pre-training their networks. Although these works remain relevant in terms of performance on ideal in-the-wild RGB data, they fail to offer a fast, self-supervised end-to-end framework capable of adapting to diverse unseen target domains (see Table 1).

# 3   Approach

We aim to accurately recover the 3D human pose for unlabeled target domain samples, while relying only on silhouette supervision to adapt a source-trained model-based regressor.

## 3.1   Background and motivation

Traditionally, weakly-supervised human mesh estimation has been framed as a bundle-adjustment problem [5]. Given the SMPL body model [38], an iterative fitting algorithm aims to adjust the pose, shape and camera parameters to minimize the reprojection loss against the given 2D observations.

**3.1.1  Model-based pose regression network.**   Given an input image $I$, a CNN regressor outputs the SMPL [38] parameters (pose $\theta$, shape $\beta$) alongside the camera parameters, $R_c = \{R, s, t\}$ (as shown in Fig. 2A). Here, $R_c$ includes global orientation $R \in \mathbb{R}^{3 \times 3}$, scale $s \in \mathbb{R}$, and translation $t \in \mathbb{R}^2$. Following this, the SMPL module generates the ordered mesh vertices in the canonical space, *i.e.* $\hat{V} \in \mathbb{R}^{6890 \times 3} = \mathcal{M}(\theta, \beta)$. The mesh vertices $\hat{V}$ are mapped to 3D pose $\hat{Y}$ (*i.e. vertex-to-3DPose*) with $k$ joints using a pre-defined linear regression: $\hat{Y} \in \mathbb{R}^{k \times 3} = W_p \hat{V}$. We can then project 3D pose $\hat{Y}$ to 2D pose $\hat{Z} \in \mathbb{R}^{k \times 2}$ using weak-perspective camera projection: $\hat{Z} = \pi(R\hat{Y}, s, t)$.

**3.1.2  2D pose supervision.**   In the presence of 2D pose ground-truth (GT), prior approaches [23, 47, 5, 33] aim to obtain the same representation for the prediction branch in order to employ a direct loss. Here, the *vertex-to-3DPose* mapping is already well specified in the SMPL module via fixed regression parameters $W_p$. Such approaches heavily benefit from this pre-specified body-joint structure. As a result, the mesh fitting problem boils down to minimizing the distance between two ordered point-sets (*i.e.* a one-to-one mapping between the 2D joint locations) which is easily achieved

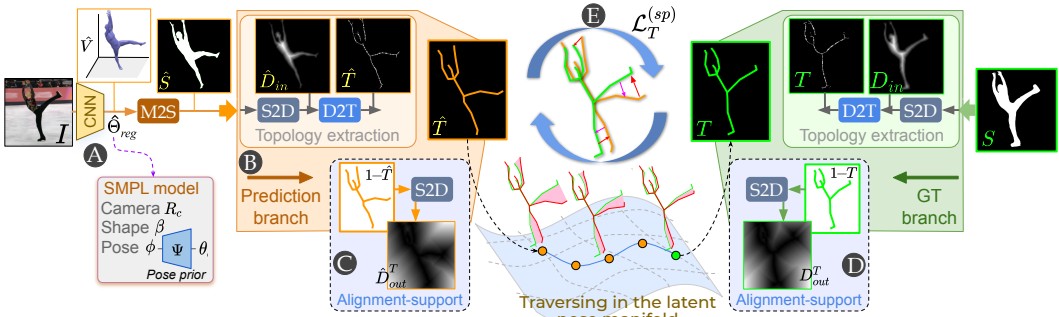

Figure 2: Framework overview highlighting the modules and the series of transformations to realize the topology alignment objective. **A.** Image to SMPL regression (Sec. 3.1.1). **B.** Extracting topological-skeleton via inwards distance-map of silhouette mask (Sec. 3.2). **C.** and **D.** Obtaining outwards distance-map of topological-skeleton to support spatial-chamfer (Sec. 3.3). **E.** Chamfer-inspired topology-alignment objective (Sec. 3.3).

via a direct $L1/L2$ loss. Note that in such cases, post SMPL-regression, all the intermediate mappings operate on pre-specified ordered point-set representations, bypassing the need for a self-sufficient articulation-centric gradient via a rendering pipeline (*i.e.* a mapping from point-set to spatial-maps).

**3.1.3 Silhouette supervision.** Unlike 2D pose supervision, learning from raw silhouette GT comes with a multitude of challenges. Though one can easily obtain the spatial silhouette from the predicted SMPL mesh via differentiable rendering [50], devising an effective self-sufficient loss stands as the major bottleneck. A trivial loss to align the two binary silhouette masks would be a pixel-level $L1$ or cross-entropy loss, as employed in general segmentation tasks. However, such a loss fails to capture any gradient information along the spatial direction (see Fig. 3A). Certain prior-arts [17, 33, 12] employ a chamfer based distance between two point-sets, termed as *fitting-loss*. Here, the two point-sets are basically the FG pixel-locations obtained from the GT and predicted silhouettes. However, this is meant for a better shape fitting and is almost always used in tandem with direct 2D or 3D pose supervision. When employed in isolation, it often lead to sub-optimal degenerate solutions, specifically in the absence of any pose-related articulation component.

Based on the above observations, we propose a new representation space, termed as *topological-skeleton*, which can facilitate seamless morphological silhouette alignment via effective spatial distance computations. To this end, we introduce the following:

**a) Distance-map, $D$.** For a given silhouette-mask $S$, its *distance-map* is defined as a spatial map $D(u)$, whose intensity at each pixel-location $u \in \mathbb{U}$ represents its distance from the closest mask-outline pixel of $S$ (see Fig. 2B). We utilize two different distance-maps: a) inwards distance map, $D_{in}$, and b) outwards distance map, $D_{out}$. The *inwards distance-map* holds zero intensity for BG pixels (*i.e.* pixels outside the active region of $S$), and vice-versa for the *outwards distance-map*. Mathematically,

$$D_{in}(u) = \begin{cases} \min_{\{u':S(u')=0\}} |u - u'|, & \text{if } S(u)=1, \\ 0, & \text{otherwise.} \end{cases} \qquad D_{out}(u) = \begin{cases} \min_{\{u':S(u')=1\}} |u - u'|, & \text{if } S(u)=0, \\ 0, & \text{otherwise.} \end{cases} \tag{1}$$

**b) Topological-skeleton, $T$.** *Topological-skeleton* is a thin-lined pattern that represents the geometric and structural core of a silhouette mask. This can also be interpreted as the ridges-lines [4, 13, 8] of $D_{in}$ extracted for a given silhouette (see Fig. 2B). For example, people with arms wide open would have the same structural topology, irrespective of their body-build type. In the absence of direct pose-centric supervision (*i.e.* 2D or 3D pose annotations), we propose to treat *topological-skeleton $T$* as a form of pose-centric influence that can be extracted from the raw silhouette. Further, we aim to realize $T$ in the form of a binary spatial map, unlike the point-set form of 2D pose keypoints. Here, $T(u)=1$ indicates that pixel $u \in \mathbb{U}$ is present in the topological ridge.

We rely on distance-map to; a) construct the topological-skeleton via inwards distance-map of the silhouette (see Fig. 2B) as discussed in Sec. 3.2.1, and b) devise a Chamfer-inspired training objective via outwards distance-map of the topological-skeleton (see Fig. 2C) as discussed in Sec. 3.3. Note that, the outwards distance-map of the topological-skeleton is defined by replacing $S$ with $T$ in Eq.1.

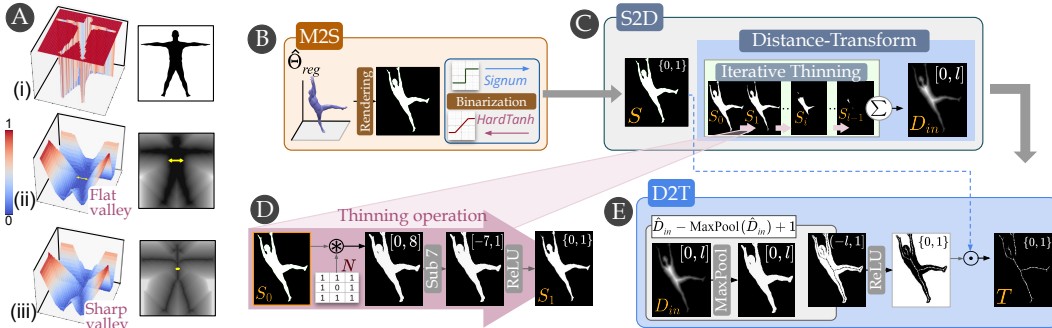

Figure 3: **A.** A representative spatial-gradient landscape for different kind of silhouette losses (left: Z-axis indicates gradient magnitude, right: intensity indicates gradient magnitude) such as, i) Pixel-level $L1$ or cross-entropy loss, ii) Chamfer inspired silhouette loss, iii) Chamfer-inspired topology-alignment loss. **B.** Details of the M2S module (see Sec. 3.2). **C.** Details of the S2D module (see Sec. 3.2.1) where **D.** shows details of a single convolutional thinning. **E.** Details of the D2T module (see Sec. 3.2.2). Pixel intensity range shown on top-right.

## 3.2 Silhouette topology extraction

Here, the primary objective is to formalize a series of transformations in order to extract the respective topological-skeletons for the predicted and GT silhouette masks, as shown in Fig. 2.

In the prediction pathway, the silhouette is obtained via a fast differentiable renderer [50] which operates on the mesh vertices $\hat{V}$. However, a non-binary silhouette is not suitable for distance-map and topology definitions as discussed in Sec. 3.1. Thus, we employ a binary activation function [18] as shown in Fig. 3B. The final outcome is represented as $\hat{S} = \texttt{M2S}(\hat{V})$. While formulating the later transformation modules, one must pay special attention to design constraints discussed below.

**Design constraints.** We intend to update the CNN by back-propagating a loss which would be computed at the topological representation, $\hat{T}$ (see Fig. 2). An ineffective regressor output $\hat{\Theta}_{reg}$ to $\hat{T}$ mapping results in vanishing gradient issues. Additionally, parallelizable operations remain critical and desirable for faster training on modern GPUs. Hence, we aim to formulate each transformation via convolution-friendly spatial operations, thereby avoiding any spatial-map to point-set mapping while implementing the topology-extraction and alignment objectives.

**3.2.1 Silhouette to distance-map, S2D.** Restricting to spatial convolution-friendly computation, we devise a recursive erosion (or thinning) operation in order to gradually shave off the boundary regions of the silhouette mask. To this end, we compute the number of active neighbours of a pixel by convolving the binary silhouette input with an $n \times n$ neighbourhood kernel $N$, as shown in Fig. 3D. The boundary pixels are expected to have less than maximum neighbours (*i.e.* $< 8$ neighbours for $n = 3$), these non-maximal pixels are then purged via the ReLU activation to output a thinned binary silhouette map, *i.e.* $S_{i+1} = \text{ReLU}(N \circledast S_i - m)$. Here, $\circledast$ denotes a convolution operator and $m = n^2 - 2$. Following this, all the series of thinned binary maps are added to realize a distance map equivalent,

$$D_{in} = \texttt{S2D}(S) = \sum_{i=0}^{l} S_i \quad \text{where:} \quad S_0 = S \quad \text{and} \quad S_{i+1} = \text{ReLU}(N \circledast S_i - m) \quad (2)$$

Interestingly, the *outwards distance-map* of $S$ can also be computed as $D_{out} = \texttt{S2D}(1 - S)$.

**3.2.2 Distance-map to topological-skeleton, D2T.** We aim to realize the thin-lined topological-skeleton as the ridge-line of the distance-map $D_{in}$. In other words, the ridge-line pixels are the local maxima of $D_{in}$. In view of convolution-friendly backpropable implementation, a pixel would be selected as a ridge-pixel if its distance-map intensity matches with the local MaxPooled map of the same (*i.e.* where $D_{in} - \text{MaxPool}(D_{in}) = 0$). Following this, the topological-skeleton is realized as:

$$T = \texttt{D2T}(D_{in}, S) = \text{ReLU}(D_{in} - Maxpool(D_{in}) + 1) \odot S \quad (3)$$

Where $\odot$ represents element-wise dot product. Note that, intensities of $D_{in}$ are discrete values in the range $[0, l]$ *i.e.*, $0, 1, 2, .., l$. Thus, the $+1$ term followed by ReLU selects the ridge-pixels along with the false-positives from the BG region, which are later pruned by masking using $S$ (see Fig. 3E). The resultant $T$ is also a binary map by formulation, i.e $T(u) \in \{0, 1\}$.

In summary, the prediction pathway involves a series of transformations, M2S→S2D→D2T to realize $\hat{V} \to \hat{S} \to \hat{D}_{in} \to \hat{T}$ (see Fig. 2). Similarly, the GT pathways involves S2D→D2T to realize $S \to D_{in} \to T$.

### 3.3 Topological loss

Here, we discuss the formulation of our topological loss, which aims to quantify the misalignment between $\hat{T}$ and $T$. We draw inspiration from Chamfer distance [11] which quantifies the misalignment between the predicted and GT point-sets. Let $T_p$ and $\hat{T}_p$ be the point-set representation of $T$ and $\hat{T}$ respectively, implying $T_p = \{u : T(u) = 1\}$. Note that, unlike 2D pose keypoints, $|T_p| \neq |\hat{T}_p|$ and there exist no point-to-point correspondence. One-way-Chamfer measures the sum of the shortest distance of each point in $T_p$ against all points in $\hat{T}_p$. Thus, the two-way Chamfer is expressed as,

$$\mathcal{L}_T^{(p)} = \mathcal{L}_{T_p \to \hat{T}_p}^{(p)} + \mathcal{L}_{\hat{T}_p \to T_p}^{(p)} = \sum_{u \in T_p} \min_{u' \in \hat{T}_p} \|u - u'\|_2^2 + \sum_{u \in \hat{T}_p} \min_{u' \in T_p} \|u - u'\|_2^2 \tag{4}$$

However, in the view of convolution-friendly implementation we aim to avoid any spatial-map to point-set mapping. Implying, we can not obtain $T_p$ or $\hat{T}_p$ as required to compute the above loss term. To this end, we propose to make use of the distance-map representation to formalize a similar loss term that would directly operate on spatial-maps. We recognize that the outwards distance-map of $T$, *i.e.* $D_{out}^T(u)$, stores the shortest distance of each BG pixel $u$ against the active pixels in $T$. Thus, an equivalent form of $\mathcal{L}_{\hat{T}_p \to T_p}^{(p)}$ can be computed as; $\|(D_{out}^T \odot \hat{T})\|_{1,1}$ where $\|.\|_{1,1}$ computes the $L1$-norm of the vectorized matrix. Thus, the final Chamfer-inspired topology-alignment loss is realized as,

$$\mathcal{L}_T^{(sp)} = \|\hat{D}_{out}^T \odot T\|_{1,1} + \|D_{out}^T \odot \hat{T}\|_{1,1} \quad \text{where: } \hat{D}_{out}^T = \text{S2D}(1-\hat{T}) \text{ and } D_{out}^T = \text{S2D}(1-T) \tag{5}$$

Note that, $\text{S2D}(1-T)$ yields the outwards distance-map (see Fig. 2) which is inline with Eq. 1. The above loss formulation effectively addresses all the design and gradient related difficulties thereby facilitating a seamless morphological silhouette alignment.

### 3.4 Adaptation training

We posit the training as a self-supervised domain adaptation problem. To this end, we first train the CNN regressor by supervising on labeled source samples, $(I, \Theta_{reg}) \in \mathcal{D}_{src}$ which is typically a graphics-based synthetic domain or an in-studio dataset. Upon deployment to an unknown target environment, we gather the image and silhouette pairs, $(I, S) \in \mathcal{D}_{tgt}$ where $S$ is obtained via classical FG-mask estimation techniques. Alongside the topology alignment objective (*i.e.* minimizing $\mathcal{L}_T^{(sp)}$), we adopt the following to regularize the adaptation process.

**a) Enforcing natural priors**. Most works rely on adversarial prior-enforcing losses to restrict the CNN predictions within the realm of natural pose and shape distributions. To simplify adaptation process, we train an adversarial auto-encoder [40, 31, 28, 48] to learn a latent pose space $\phi \in [-1,1]^{32}$ whose decoder, $\Psi$ generates plausible 3D pose vectors $\theta \in \mathbb{R}^{3k} = \Psi(\phi)$. Thus, just integrating this frozen decoder into our framework constrains the pose to vary within the plausibility limits. We denote $\hat{\Theta}_{reg} = \{\beta, \phi, R_c\}$ and regularize $\beta$ to remain close to the mean shape, inline with [26].

**b) Optimization in the loop.** Motivated by the benefits of combining regression and optimization based training routines [26], we propose a target adaptation procedure as shown in Algo. 1. We first initialize $\Theta_{reg}^{(opt)}$ by inferring $\hat{\Theta}_{reg}$ from the current state of the source-trained CNN. These regression-initialized $\Theta_{reg}^{(opt)}$ parameters undergo an iterative fitting procedure which aims to minimize $\mathcal{L}_T^{(sp)}$ and $\mathcal{L}_S^{(sp)}$ without updating the CNN parameters. Here, $\mathcal{L}_S^{(sp)}$ represents the spatial chamfer-based loss directly applied on raw-silhouettes, obtained

---

$\Theta_{CNN}$: Source training initialized CNN weights
$\Theta_{reg}^{(opt)}$: SMPL parameters initialized from $\hat{\Theta}_{reg}$
**for** *iter < MaxIter* **do**
    **if** *iter* $(\text{mod } K) \neq 0$ **then**
        Update $\Theta_{CNN}$ by optimizing $\mathcal{L}_S^{(sp)}, \mathcal{L}_T^{(sp)}$.
    **else**
        **for** *iter_opt < MaxIter_opt* **do**
            Fit $\Theta_{reg}^{(opt)}$ to minimize $\mathcal{L}_S^{(sp)}, \mathcal{L}_T^{(sp)}$
        Update $\Theta_{CNN}$ by optimizing $\mathcal{L}_\Theta$.

**Algorithm 1:** Proposed adaptation procedure.

---

by replacing $T$ and $\hat{T}$ with $S$ and $\hat{S}$ in Eq. 5. This loss further improves pose and shape fitting. Subsequently, the fitted $\Theta_{reg}^{(opt)}$ is used to form a supervised loss, $\mathcal{L}_\Theta = \|\hat{\Theta}_{reg} - \Theta_{reg}^{(opt)}\|$ to update the CNN weights, $\Theta_{CNN}$. Although, we primarily rely on the end-to-end training of the CNN regressor by directly minimizing the proposed alignment objectives, alternating between regression and fitting based routines allows the model to self-improve faster.

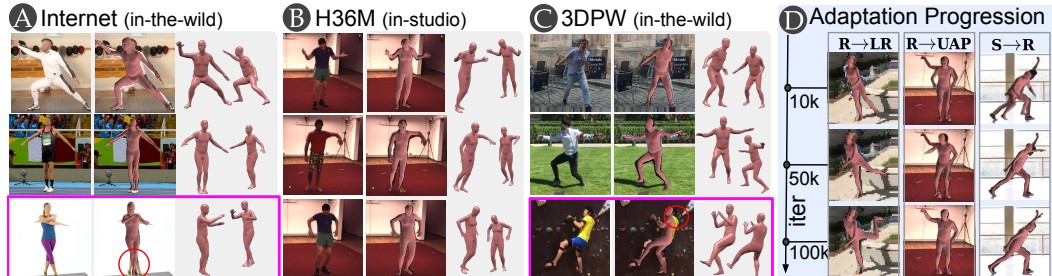

Figure 4: **A**, **B**, and **C**. Qualitative results of *Ours(S→R)* on the respective datasets. Failure cases are highlighted in magenta. **D**. Progression of our articulation-centric adaptation across various domains (across columns).

Table 2: Comparison against prior-works on Human3.6M (P-2). Table is divided based on access to supervision. We beat prior arts at comparable supervision levels.

| Sup. | Method | PA-MPJPE($\downarrow$) |
|---|---|---|
| Full | Pavlakos *et al.* [47] | 75.9 |
| | HMR [23] | 56.8 |
| | SPIN [26] | 41.1 |
| Weak | HMR (unpaired) [23] | 66.5 |
| | SPIN (unpaired) [26] | 62.0 |
| | *Ours(S→R,weak)* | **58.1** |
| Unsup. | Kundu *et al.* (unsup) [29] | 90.5 |
| | *Ours(S→R)* | **81.3** |

Table 3: Evaluation on the 3DPW dataset, none of the works train on 3DPW. For fair comparison, we separate works based on access to target supervision.

| Sup. | Method | MPJPE($\downarrow$) | PA-MPJPE($\downarrow$) |
|---|---|---|---|
| Full | HMR [23] | 128.1 | 81.3 |
| | Kanazawa *et al.* [24] | 116.5 | 72.6 |
| | SPIN [26] | 98.6 | 59.2 |
| Weak | Martinez *et al.* [41] | - | 157.0 |
| | SMPLify [5] | 199.2 | 106.1 |
| | Doersch *et al.* (RGB+2D) [10] | - | 82.4 |
| | *Ours(S→R, weak)* | **126.3** | **79.1** |
| Unsup. | Doersch *et al.* (DANN) [10] | - | 103.0 |
| | Kundu *et al.* (unsup) [29] | 187.1 | 102.7 |
| | Doersch *et al.* (Flow) [10] | - | 100.1 |
| | *Ours(S→R)* | **159.0** | **95.1** |

# 4 Experiments

We perform a thorough empirical analysis demonstrating our superiority for cross domain adaptation.

**Implementation details.** We use an ImageNet [52] pre-trained *ResNet-50* [16] network as our CNN backbone for the regressor. The final layer features are average pooled and subsequently passed through a series of fully-connected layers to regress the latent pose encoding $\phi$, shape and camera parameters. We use the Adam optimizer [25] with a learning rate $1e{-}6$ and batch size of 16, while setting *MaxIter*$_{opt}$ to 10 and $K{=}4$. We use separate optimizers for $\mathcal{L}_T^{(sp)}$ and $\mathcal{L}_S^{(sp)}$. A single iteration of the iterative fitting procedure takes nearly 12ms on a Titan-RTX GPU.

**Datasets.** Moshed [37] CMU-MoCap [32] and H3.6M training-set [19] form our unpaired 3D pose data, which is used to train the 3D pose-prior. Our adaptation settings use the following datasets.

**a) Synthetic (S):** We use SURREAL [57] to train our synthetic source model. It is a large-scale dataset with synthetically generated images of humans, rendered from 3D pose sequences of [32].

**b) Real (R):** We use a mixture of Human3.6M [19] and MPII [1] as our Real-domain dataset. This domain is used as both source and target in different adaptation settings.

**c) UAP-H3M (UAP):** Universal Adversarial Perturbation (UAP) [44] is an instance-agnostic perturbation aimed at attacking a deployed model, thereby inflicting a drop in the task performance. Typically, the adversarial noise is added with the clean source domain images to construct the adversarially perturbed target domain samples. We craft a single UAP perturbation ($L_\infty$ and $\epsilon{=}8/255$) using an equivalent 3D pose estimation network while following the procedure from [45]. We aim to evaluate the effectiveness of the proposed self-supervised adaptation as a defence mechanism against such perturbed domains. We perturb the clean Human3.6M [19] samples to construct such a target domain, denoted as `UAP-H3M`. Further, we experiment on three perturbation strengths while varying $\epsilon$ as $4/255$, $8/255$, and $16/255$ (see Table 5).

**d) LR-3DPW (LR):** We use low-resolution (LR) variants of 3DPW [58] dataset as another target domain, denoted as `LR-3DPW`. LR holds practical significance as they are common in surveillance and real-time streaming applications. Several works [46, 60] acknowledge the performance drop in models trained on high-resolution (HR) data, while evaluating on LR targets. Addressing this, Xu *et al.* [60] construct an LR-specific architecture and assume simultaneous access to pose-labeled samples from both LR and HR domains, hence not comparable in our setting. We aim to evaluate the proposed self-supervised adaptation technique as a remedy to such domain-shifts in the absence of target

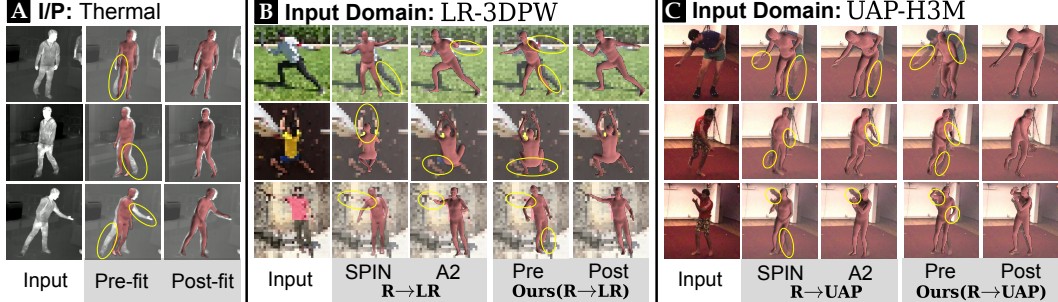

Figure 5: **A.** Inference-time fitting on Thermal images. **B.** Results on LR–3DPW) **C.** Results on UAP–H3M). Yellow ellipses highlight the region of articulation errors (refer Table 5 caption for comparison).

domain pose-labels. We construct three different LR domains by downsampling the full-resolution image to $w \times w$ where $w$ is set as 96, 52 and 32 (see Table 5). Each LR domain image is upsampled (bi-cubic) back to $224 \times 224$ before forwarding them through the CNN regressor.

**Adaptation settings.** We evaluate our approach on the following adaptation settings. We use full supervision for source pre-training and use only silhouettes for target adaptation. In static-camera moving-object scenarios, silhouettes obtained via BG subtraction remain unaffected by domain shifts.

**a)** *Ours(S→R)* and *Ours(S→R, weak).* In this setting, we initialize our Synthetic source model by accessing full supervision from SURREAL [57] and undertake topological-alignment based adaptation on the *Real* domain. *Ours(S→R)* does not access any *Real* 2D/3D pose annotation. However, the weakly-supervised variant, *Ours(S→R, weak)*, utilizes 2D pose supervision.

**b)** *Ours(R→UAP)* and *Ours(R→LR).* Here, we adapt the *Real* domain source model, *Ours(R)*, to the UAP–H3M and LR–3DPW domains respectively, by only accessing silhouettes from the target domains.

## 4.1 Ablative study

We define three baseline models that use various silhouette-based losses for self-supervised adaptation. In Table 4, *Ours(S)* reports the pre-adaptation performance in order to gauge the corresponding post-adaption improvements. *B1*, a model adapted using only the pixel-level $L2$ loss (*i.e.* $\|S - \hat{S}\|_2^2$) shows very poor performance. This shows that $L2$ based losses fall short of offering an independent articulation-centric objective. On the other hand, replacing it with our spatial chamfer-based alignment loss $\mathcal{L}_S^{(sp)}$ on raw-silhouettes, gives reasonable adaptation performance, shown as *B2*.

Table 4: Ablation experiments on 3DPW for S→R. *B1*, *B2*, and *B3* are adaptation baselines that use different loss variants as listed below.

| Method | Loss on $S$ | Loss on $T$ | PA-MPJPE ($\downarrow$) |
|---|---|---|---|
| *Ours(S)* | - | - | 134.7 |
| *B1(S→R)* | as $L2$ | - | 129.6 |
| *B2(S→R)* | $\mathcal{L}_S^{(sp)}$ | - | 106.2 |
| *B3(S→R)* | $\mathcal{L}_S^{(sp)}$ | as $L2$ | 105.5 |
| *Ours(S→R)* | $\mathcal{L}_S^{(sp)}$ | $\mathcal{L}_T^{(sp)}$ | **95.1** |

We observe that naively supplementing *B2* with an $L2$ loss on topological-skeleton (*i.e.* $\|T - \hat{T}\|_2^2$) yields very minimal benefit, despite $T$ containing useful articulation-centric information. Finally, we compare these baselines with our proposed model to clearly establish the effectiveness of our Chamfer inspired topological-alignment loss towards driving articulation-centric learning.

## 4.2 Comparison against prior works

Adhering to standard metrics, we compute the mean per joint position error (MPJPE) [19] and Procrustes aligned [14] mean per joint position error (PA-MPJPE) to evaluate the pose specific adaptation performance. We evaluate on Human3.6M [19] following Protocol-2 [23].

**a) Adaptation from Synthetic to Real.** We evaluate the proposed approach against prior human mesh recovery works on both Human3.6M [19] (Protocol-2) and 3DPW [58] datasets. Our model achieves *state-of-the-art* performance on self-supervised and weakly-supervised settings (see Table 2 and Table 3). Though we access synthetic domain data similar to Doersch *et al*. [10], we deem ourselves comparable as other listed works [23, 26, 47, 24] benefit from access to additional *Real* domain datasets such as LSP, LSP-Extended [22] and COCO [35].

**b) Adaptation from Real to UAP-H3M.** We obtain pre-adaptation performance of the source trained networks via direct inference on the shifted target (first 2 rows of Table 5). We report these in order to gauge the severity of the domain-gap. In *A1*, SPIN [26] is finetuned on UAP–H3M targets by using 2D pose predictions from off-the-shelf OpenPose [7, 6, 59] network. Only the high-confidence 2D

Table 5: Evaluation on UAP-H3M (P2) and LR-3DPW. We compare against two strong adaptation baselines: using 1) $A1$: 2D pose from an off-the-shelf network [6], 2) $A2$: additional point-set based silhouette fitting-loss [17]. Comparing $A1$, $A2$, and *Ours* on pre-to-post performance recovery, we observe that the effectiveness of our approach increases with increasing domain shift. + represents adaptation using the specified loss.

| | Method | Adaptation from R to UAP-H3M | | | | | | Adaptation from R to LR-3DPW | | | | | |
| | | MPJPE (↓) | | | PA-MPJPE (↓) | | | MPJPE (↓) | | | PA-MPJPE (↓) | | |
| | | 4/255 | 8/255 | 16/255 | 4/255 | 8/255 | 16/255 | 96 | 52 | 32 | 96 | 52 | 32 |
|---|---|---|---|---|---|---|---|---|---|---|---|---|---|
| Pre-Adapt. | SPIN [26] | 65.8 | 98.2 | 160.1 | 44.6 | 60.8 | 90.7 | 104.3 | 120.3 | 176.4 | 63.7 | 71.1 | 87.9 |
| | *Ours(R)* | 67.7 | 103.9 | 161.8 | 46.9 | 63.6 | 91.2 | 110.8 | 127.5 | 178.1 | 68.6 | 76.3 | 88.2 |
| Post-Adapt. | $A1$: SPIN+$\mathcal{L}_{2D}^{(p)}$ | 64.5 | 94.0 | 151.2 | 43.4 | 59.5 | 89.8 | 100.2 | 117.0 | 153.6 | 61.7 | 70.3 | 85.4 |
| | $A2$: SPIN+$\mathcal{L}_{2D}^{(p)}$+$\mathcal{L}_{S}^{(p)}$ | 64.1 | 89.1 | 136.5 | 43.4 | 58.9 | 85.1 | 100.1 | 115.2 | 147.5 | 61.5 | 69.8 | 82.3 |
| | *Ours(R→UAP)* | **63.6** | **84.7** | **125.2** | **43.2** | **57.6** | **79.4** | - | - | - | - | - | - |
| | *Ours(R→LR)* | - | - | - | - | - | - | **99.8** | **114.7** | **134.2** | **61.3** | **68.7** | **78.9** |

predictions are used, inline with [26]. To create a strong baseline, in $A2$, we supplement the $A1$ setting with additional silhouette supervision, via the *fitting-loss* [17]. As shown in last two rows of Table 5, the proposed adaptation method clearly imparts superior recovery of target performance, despite accessing only silhouettes. The baselines $A1$ and $A2$, fail to recover from domain shift despite having access to stronger supervision. We observe that, unlike the silhouette extraction process, OpenPose itself suffers from domain shift issues, thus is ineffective towards improving target performance.

**c) Adaptation from Real to LR-3DPW.** Here, we undertake adaptation to the LR-3DPW domain and evaluate against the same baseline settings as in R→UAP task. But unlike in UAP, the target silhouettes here are of lower resolution. Despite this, in Table 5, we clearly see superior adaptation performance against the baselines ($A1$ and $A2$) which access additional 2D-pose predictions obtained via OpenPose [7, 6, 59]. These observations shine light on the ineffectiveness of network-based pseudo supervision (via OpenPose) towards recovering from domain-shift issues, and voices the need and usefulness of our framework in practical deployment scenarios.

**Qualitative results.** Fig. 5 shows qualitative results on LR-3DPW and UAP-H3M. Interestingly, our formulation of fitting in-the-loop with silhouette based objectives allows us to fit meshes using only silhouettes. Hence enabling us to perform fast inference-time fitting on diverse unseen domains such as thermal images, without requiring any 2D/3D pose annotations. We show a few such fittings in Fig. 5A. We also observe that in contemporary CNN-based approaches, domain-shift primarily manifests articulation degradation, while person-orientation remains relatively robust. Additionally, in Fig. 4, we show a few pose recovery results of *Ours(S→R)* on Human3.6, 3DPW and other in-the-wild images. In Fig. 4D, we visualize the gradual improvement in the predicted mesh output across intermediate iterations of the training.

**Limitations.** Silhouettes offer significantly weaker supervision than 2D/3D pose and hence the model fails in presence of multi-level inter-part occlusions. Some complex poses also elicit limb-swapped articulations (shown in magenta in Fig. 4). Though, an only fitting based approach severely fails to solve such cases, optimization coupled with CNN training fairly recovers this deficiency as a result of the inductive bias instilled in the source trained model. Here, the inductive bias can be improved by training on multi-source data, such as combining $S$ and $R$ for adapting to extreme domain shifts like thermal or NIR. In future, it remains interesting to explore ways to address such deficiencies via temporal and multi-frame articulation consistencies.

**Negative societal impacts.** Although our approach in itself does not pose any direct negative societal impacts, we understand that the task at hand (*i.e.* 3D human pose recovery) may be directed towards unwarranted and intrusive human activity surveillance. We strongly urge the practitioner to confine the applicability of the proposed method to strictly ethical and legal use-cases.

# 5 Conclusion

In this paper, we introduce a self-supervised domain adaptation framework that relies only on silhouette supervision. The proposed convolution friendly spatial distance field computation helps us disentangle the topological-skeleton, thereby facilitating seamless morphological silhouette alignment. The empirical evaluations clearly highlight the usefulness of the framework for practical deployment scenarios, even as a remedy to adversarial attack. Integrating temporal aspects, such as optical-flow, in our framework remains an interesting future direction.

## Acknowledgments and Disclosure of Funding

This work was supported by a Qualcomm Innovation Fellowship (Jogendra) and a project grant from MeitY (No.4(16)/2019-ITEA), Govt. of India. We would also like to thank the anonymous reviewers for their valuable suggestions.

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
