# Supplementary: Aligning Silhouette Topology for Self-Adaptive 3D Human Pose Recovery

In this supplementary, we provide details of the 3D pose prior module followed by additional implementation details and qualitative results.We organize this supplementary document as follows:

- Section 1: Notations and compute details
- Section 2: Instilling 3D pose prior
- Section 3: Additional qualitative results and insights

## 1 Notations and compute details

For implementation we use the PyTorch [13] framework and train our networks on a machine with NVIDIA TITAN RTX GPU (24 GB VRAM) and Intel i7-8700 CPU (48GB RAM). The source model training on an average took 2-3 days to train, while the adaptation training varied by target dataset size. A batched inference through the model takes around 20 milli-seconds. Table 1 briefly describes and lists all the notations used in the main paper.

## 2 Instilling 3D pose prior

Typically pose priors [6, 2, 8] are imposed via an adversarial discriminator where the discriminator is trained to discriminate between the real and regressor predicted poses. In such approaches, infusing priors into the regressor requires jointly optimizing the primary learning objectives alongside the cumbersome adversarial training. Despite this, the regressor may still hold the possibility of outputting implausible poses when faced with unseen data. In the context of self-supervised adaptation, using such prior infusion techniques makes the optimization highly unstable. Acknowledging this, we aim to implement a 3D pose prior network, wherein constraining the 3D pose within natural plausible distribution would just require integrating the frozen network into the framework.

We comprehend this 3D pose prior using an Adversarial Auto Encoder (AAE) [12], and train on unpaired 3D pose data from Moshed [10] CMU-MoCap [9] and Human3.6M training-set [5]. The AAE learns a bottleneck pose representation while regularizing it to follow a predefined uniform latent distribution $\mathcal{U}[-1, 1]^{32}$. The AAE setup consists of an encoder $\Phi$, a decoder $\Psi$, and an adversarial discriminator ***Disc***, as shown in Fig. 1. The latent pose encoding denoted as $\phi$, is a 32 dimensional vector.Here, the decoder $\Psi$ functions as the generative model of 3D human pose, mapping all latent codes $\phi_{sam} \sim \mathcal{U}[-1, 1]^{32}$ to plausible 3D pose articulations.

The question that arises is how to integrate this prior modeling into the CNN regressor framework. In order to enforce plausible regressor predictions, one has to simply regress the latent pose representation $\phi$ during training. Note that, a simple *tanh* non-linearity at the regressor output ensures that the regressed latent code follows $\mathcal{U}[-1, 1]^{32}$, which would always decode to plausible pose articulations. Such a formulation is beneficial as it can generate realistic (plausible) 3D pose predictions without relying on any explicit adversarial training during the self-supervised adaptation.

The following subsections provide more details:

**a) $\theta$: Parent-relative local representation.** SMPL [11] pose parameter $\theta \in \mathbb{R}^{3k}$ expresses each body joint as rotations with respect to its parent joint (*i.e.* a parent-relative local coordinate system), and towards this, the axis-angle representation is used. The axis–angle representation compactly parameterizes the 3D rotation by only two quantities: 1) a unit vector indicating the direction of an axis of rotation, and 2) an angle denoting the magnitude of rotation about that axis. The direction of rotation (*i.e.* clockwise or anti-clockwise) is given by the mnemonic of right-hand rule. This acts as the input data representation in the AAE setup. Note that unlike 3D pose skeleton, $\theta$ also contains joint rotation information. For example, twisting of wrist would yield a different $\theta$ vector depending on the amount of axial rotation.

35th Conference on Neural Information Processing Systems (NeurIPS 2021).

Table 1: Notations used in the paper.

| | Symbol | Description |
|---|---|---|
| **SMPL Related** | $\theta$ | Canonical SMPL pose |
| | $\beta$ | SMPL shape parameter |
| | $\mathcal{M}$ | SMPL module function |
| | $R_c$ | Global orientation and camera parameters |
| | $\hat{V}$ | 3D Vertex locations |
| | $\hat{Y}$ | 3D Joint locations |
| | $W_p$ | Frozen Vertex-to-joint regressor |
| | $\hat{Z}$ | Camera projected 2D joint locations |
| **3D Pose prior** | $\{\Phi, \Psi\}$ | Encoder-decoder pair of AAE |
| | $\phi$ | Latent pose vector |
| | $Disc$ | Pose discriminator used to train AAE |
| **Transform -ations** | $\pi$ | Weak perspective camera projection |
| | M2S | Mesh to binary silhouette $S$ via differentiable rendering |
| | S2D | Silhouette to distance-map $D$ transformation |
| | D2T | Distance-map to topological-skeleton $T$ transformation |
| **Representations (space and samples)** | $I$ | Input image |
| | $\mathbb{U}$ | Image coordinate space |
| | $S$ | Foreground silhouette |
| | $D_{in}$ | Inwards distance-map of $S$ |
| | $D_{out}$ | Outwards distance-map of $S$ |
| | $T$ | Topological skeleton |
| | $D_{out}^{T}$ | Outwards distance-map of $T$ |
| | $\hat{\Theta}_{reg}$ | Regressor predictions including $\phi$, $\beta$, and $R_c$ |
| | $\Theta_{reg}^{(opt)}$ | In the loop optimized regressor predictions |
| | $N$ | Convolutional neighbourhood kernel used in S2D module |
| **Scalars** | $k$ | Number of body joints |
| | $u$ | Index over image coordinate space |
| | $l$ | Number of thinning iterations |
| **Losses** | $\mathcal{L}_S^{(sp)}$ | Proposed spatial alignment loss on raw silhouettes |
| | $\mathcal{L}_T^{(sp)}$ | Proposed spatial alignment loss on topological-skeleton |
| | $\mathcal{L}_{2D}^{(p)}$ | Point-set based 2D pose loss |
| | $\mathcal{L}_S^{(p)}$ | Point-set based silhouette fitting loss |
| | $\mathcal{L}_\Theta$ | Loss on $\hat{\Theta}_{reg}$ |
| **Others** | $\mathcal{D}_{src}$ | Source domain distribution |
| | $\mathcal{D}_{tgt}$ | Target domain distribution |
| | UAP-H3M | Universal Adversary Perturbed Human3.6M [5] |
| | LR-3DPW | Low-resolution 3DPW [14] |

**b) Architecture details.** The AAE architecture consists of two Fully-Connected (FC) layers with 1024 neurons each, followed by a 512 neuron FC layer, which finally outputs a 32 dimension latent representation. We employ a symmetric encoder-decoder architecture *i.e.* 512, 1024, 1024 neuron FC layers reconstructing back the input $\theta$ vector. The latent pose encoding $\phi$, is a 32 dimensional vector and is obtained through *tanh* non-linearity. For *Disc*, $\theta$ is forwarded through a common embedding network of two fully-connected layers with 32 hidden neurons each. The outputs are then passed to $k = 23$ different discriminators, which output 1D values. A global discriminator focusing on overall pose plausibility is also employed, with two fully-connected layers of 1024 neurons each, this is finally concatenated with the $k$ outputs. All fully-connected layers use ReLU activations except the final layer, similar to the pose discriminator formulation of HMR [6].

**c) Training the pose prior.** We train the AAE with an aim to learn a pose embedding that is smooth, continuous and plausible in pose space. In order to enforce the learning of an one-to-one mapping in a generative adversarial setup, we add cyclic reconstruction loss on both input pose $\theta$ and latent pose $\phi$ as follows:

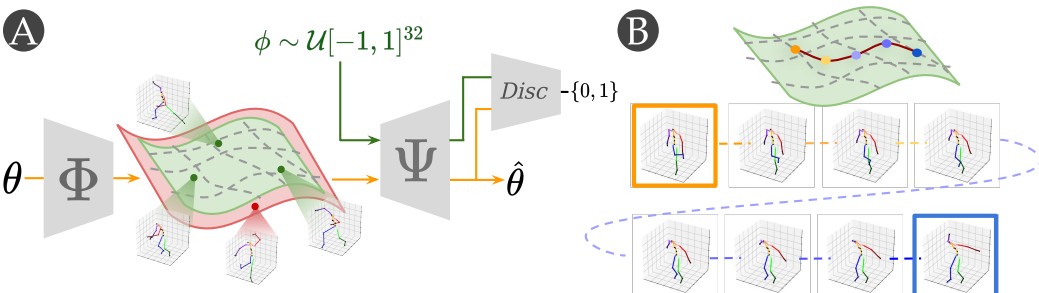

Figure 1: **A.** Training setup of the AAE based 3D Pose prior. It decodes a plausible pose when sampled in $\mathcal{U}[-1,1]^{32}$ (green region) while sampling outside this bound may lead to implausible poses (red region). **B.** Latent manifold traversal with smooth interpolative transition between poses.

$$\mathcal{L}_{cyc} = |\widehat{\theta} - \theta| + |\widehat{\phi} - \phi| \quad \text{Where:} \quad \widehat{\theta} = \Psi \circ \Phi(\theta) \quad \text{and} \quad \widehat{\phi} = \Phi \circ \Psi(\phi) \tag{1}$$

Here, $\hat{\theta}$ represents the reconstructed output for the corresponding input $\theta$. Similarly, $\phi \in [-1,1]^{32}$ and $\hat{\phi}$ are the latent pose and reconstructed latent pose of the autoencoder for a given $\theta$.

On the adversarial training side, we train the discriminator ***Disc*** to distinguish between real latent pose $\phi_{real}$ and latent pose sampled as $\phi_{sam} \sim \mathcal{U}[-1,1]^{32}$. Such an adversarial training setup ensures that the entire sampled space *i.e.* $\mathcal{U}[-1,1]^{32}$ is plausible, smooth and continuous as shown in Fig. 1. The adversarial loss function can be written as:

$$\min\ \mathcal{L}_{adv}(\Psi) = \mathbb{E}_{\phi_{sam}}[(Disc(\Psi(\phi_{sam})) - 1)^2] \tag{2}$$

$$\min\ \mathcal{L}_{adv}(Disc) = \mathbb{E}_{\phi_{real}}[(Disc(\Psi(\phi_{real})) - 1)^2] + \mathbb{E}_{\phi_{sam}}[(Disc(\Psi(\phi_{sam})))^2] \tag{3}$$

Here, the Eq. 2 denotes generative adversarial loss on decoder $\Psi$ and the Eq. 3 specifies the loss on the pose discriminator *Disc*. We train encoder $\Phi$ using $\mathcal{L}_{cyc}$ and decoder $\Psi$ using $\mathcal{L}_{cyc} + \lambda\mathcal{L}_{adv}$. Here $\lambda$ denotes the weight that steers the influence exerted by the loss terms. We empirically find that an annealing scheme for $\lambda$ helps training. Note that the Discriminator is jointly trained with $\Phi$ and $\Psi$ with the aforementioned losses. We use Adam solver [7] to optimize the losses.

## 3 Additional qualitative results and insights

**a) Qualitative results on *Real* domain datasets.** We present qualitative results of *Ours(S→R, weak)* model in Fig. 2. It consists evaluation on 3DPW [14], Human3.6 [5], MPII [1] and other in-the-wild datasets. We show mesh overlays as well as their corresponding multi-view visualisations. Failure cases are highlighted in magenta. Fig. 2 clearly highlights generalized performance of our model. The failure case in the middle column shows an erroneous prediction on a yoga posture. The model fails in such complex postures as they are rarely encountered during training. The one on the right (last row, rightmost column) shows a failure case on an acrobatic pose. We attribute the poor prediction to limb depth ambiguity.

**b) Qualitative comparison on `UAP-H3M` and `LR-3DPW`.** We show additional qualitative comparisons against SPIN [8] and the baseline *A2* (defined at Sec. 4.2 of main paper ) in addition to comparing our pre and post adaptatation networks. Fig. 3A shows articulation centric comparison on the low-resolution 3DPW [14], `LR-3DPW`, and Fig. 3B shows articulation centric comparison on the Univeral adversary perturbed Human3.6 [5], `UAP-H3M`, datasets (defined in Sec. 4 of main paper). Yellow ellipses highlight the region of articulation errors. We obtain pre-adaptation results of the source trained networks via direct inference on the shifted target (column 2 and 4 in panel A and B of Fig 3). We show these in order to qualitatively compare the improvement. We observe that the proposed adaptation method imparts superior recovery in target performance, despite accessing only silhouettes. The strong baseline of *A2*, fails to recover from domain shift despite having access to stronger supervision (see column 2 and 3 of Fig. 3). We observe that, OpenPose [4, 3, 15] itself suffers from domain shift issues, and thus is ineffective towards improving target performance.

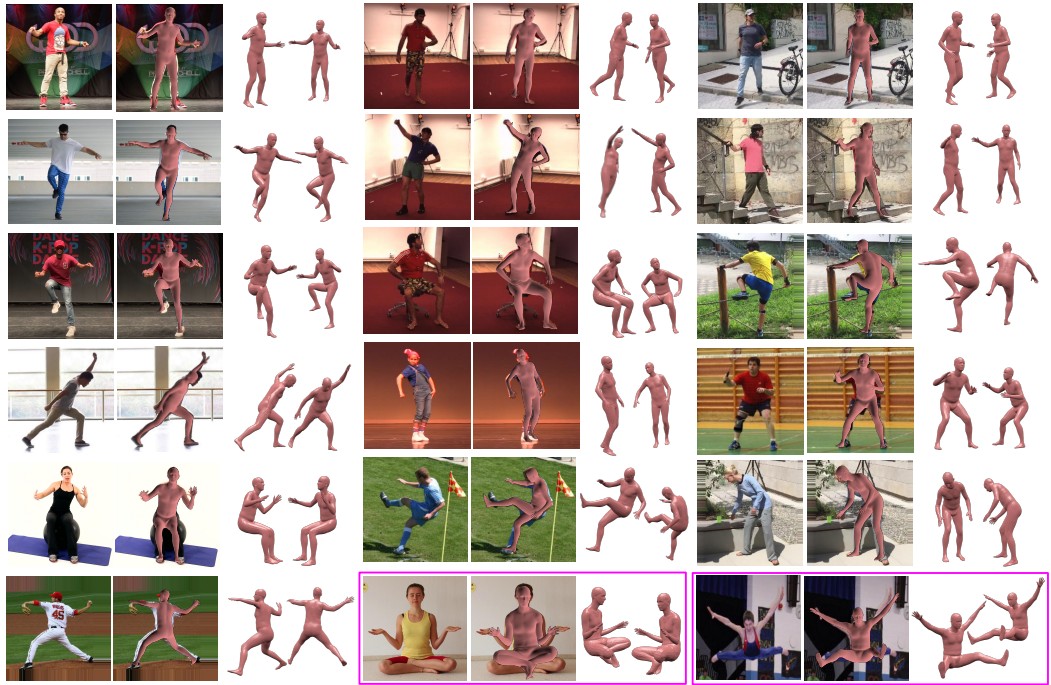

Figure 2: Qualitative results of *Ours(S→R, weak)* on various datasets. Failure cases are highlighted in magenta (see Sec. 3 for details).

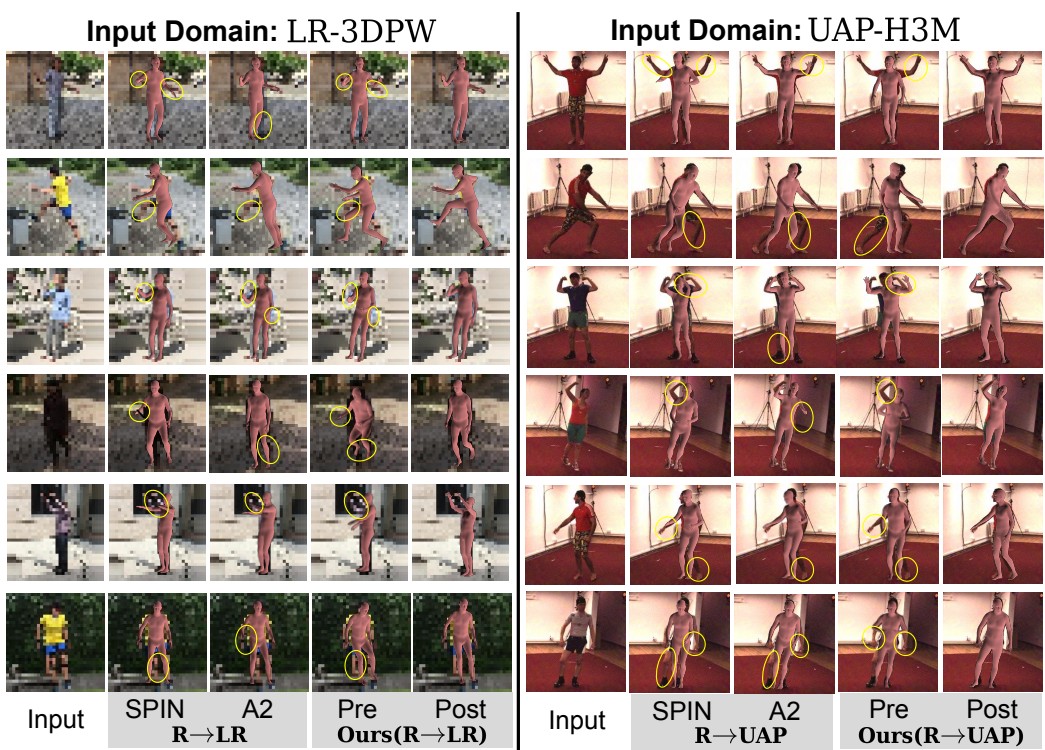

Figure 3: Qualitative comparison against baseline *A*2 on `LR-3DPW` and `UAP-H3M` (see Sec. 3 for details). The proposed adaptation method imparts superior recovery in target performance, despite accessing only silhouettes. Yellow ellipses highlight the region of articulation errors.

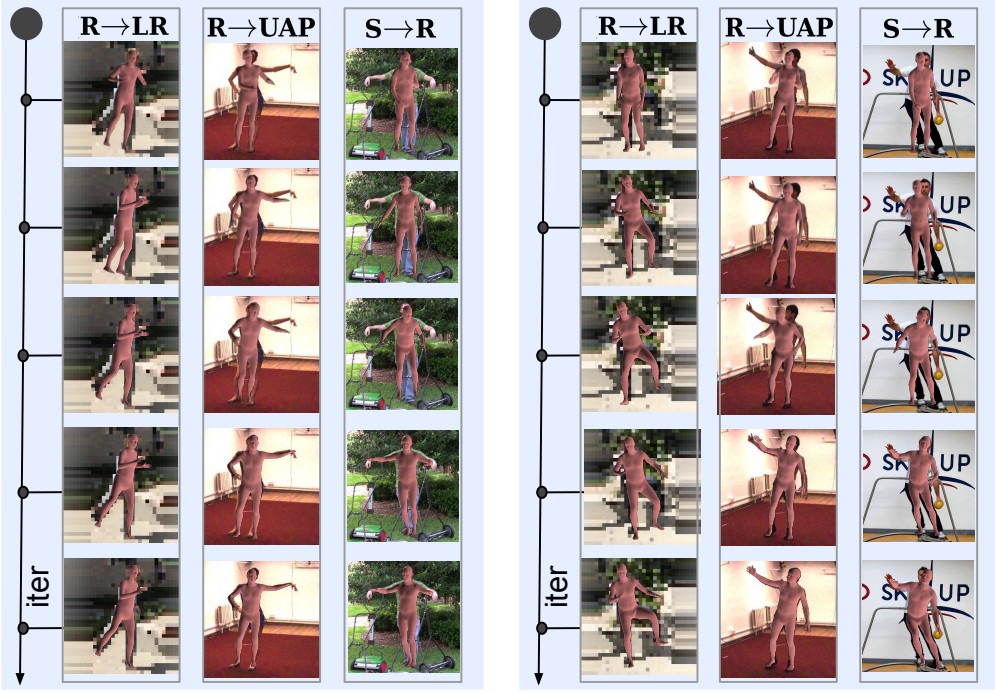

Figure 4: Progression of our articulation-centric adaptation across various domains (across columns). The gradual improvement in target prediction conveys the effectiveness of the proposed framework.

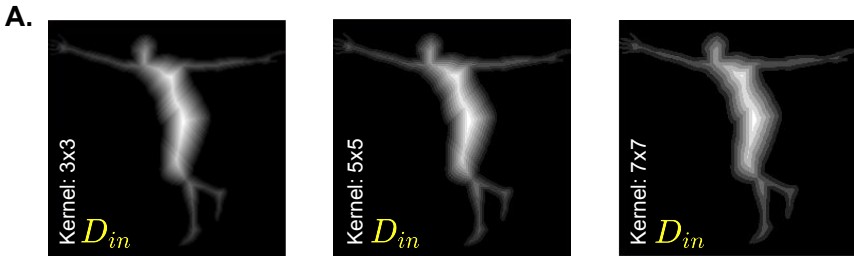

Figure 5: With increase in the kernel size ($3\times3$, $5\times5$, and $7\times7$), S2D module outputs distance maps $D$ with increasing intensity-quantization.

Table 2: A quantitative comparison of the proposed S2D module against various distance map definitions. Kernel size here refers to the kernel size used by the proposed approach. The proposed approach is closer in approximation to Chebyshev (or chessboard) distance map computation.

| Comparison Vs. | Kernel size | SSIM | PSNR (dB) |
|---|---|---|---|
| Exact Euclidean distance map | $3\times3$ | 0.981 | 31.4 |
| Taxicab distance map | $3\times3$ | 0.961 | 24.3 |
| Chebyshev distance map | $3\times3$ | 0.996 | 75.8 |
| Chebyshev distance map | $5\times5$ | 0.930 | 17.9 |
| Chebyshev distance map | $7\times7$ | 0.882 | 12.3 |

**c) Visualizing adaptation progression.** In Fig. 4, we visualize the gradual improvement in the predicted mesh output across intermediate iterations of the training. We show several sequences on $R{\rightarrow}$UAP-H3M, $R{\rightarrow}$LR-3DPW and $S{\rightarrow}R$ adaptation settings (see Sec. 4 of main paper).

**d) Comparing** S2D **module against other distance map definitions.** With increase in the kernel size of N ($3{\times}3$, $5{\times}5$, and $7{\times}7$), S2D outputs distance maps $D$ with increasing intensity-quantization error against the true distance map (refer Fig.5). In our experiments, we use $3{\times}3$ kernel size as it is able to resolve and account for sharp deviations (such as at object corners or at thin foreground regions). Table 2 reports the quantitative comparison of the proposed approach against various distance map definitions. The quantitative analysis shows that the proposed approach is closer in approximation to Chebyshev (or chessboard) distance map computation, primarily due to its convolutional nature.