# OpenReview forum: "Aligning Silhouette Topology for Self-Adaptive 3D Human Pose Recovery"
_NeurIPS.cc/2021/Conference — NeurIPS 2021 Poster_

### Official Review · Reviewer_xChc · 2021-07-16

**Rating:** 7
**Confidence:** 4

**Summary:**

This work proposes a novel method to tackle the lack of annotated data for 3D human pose recovery. The core observation is that silhouette estimation is not prone to domain gaps. Hence, the authors develop the topological skeleton, obtained from silhouette, to guide the training.

**Limitations And Societal Impact:**

The authors have stated clearly the limitations and societal impact, and provide sound solutions to address them.

**Main Review:**

Strengths:
- The paper is well-written. It is a joy reading this paper.

- The proposed method is sound and interesting. Specifically, 1) transferring chamfer distance, a normally 3D loss to images is innovative; 2)  computation of the ridge-lines via convolution-friendly operations is carefully designed and beautifully implemented.

- Thorough ablation studies.

- Insightful discussions. The observation on the robustness of silhouette estimation across domains is highly thought-provoking. Also, it is less relevant but important to highlight that articulation degradation is the key result of the domain gap (Line 345).

Weaknesses:
-  Please explain the setting used for HMR and SPIN in Table 2, under the weak sup. setting. What does "unpaired" mean?

- Optimization in the loop is used. However, this involves using a well-established regressor. It raises a concern that the network regressor may play an over-important role in determining the performance of the method, because the regressor is trained in a supervised way, usually making it much better even if there exist domain gaps.

- As stated in the Limitation section that "inter-part occlusion" remains a challenge, is this an inherent limitation of the proposed method? i.e. impossible to solve?

- Unlike many fields in which unsupervised learning can produce close results to the supervised learning counterpart, the unsupervised learning in 3D human pose recovery is, however, far from being practical since there is a large performance gap. I would like to hear the authors' comments on this.

Minor:
- Line 332: it may be more convincing to provide some qualitative evidence to support the statement.

**Time Spent Reviewing:**

6

---

> ### Author Response · Authors · 2021-08-10
> **Response to Reviewer xChc**
>
> We thank the reviewer for a succinct review. We are encouraged to hear that the reviewer enjoyed reading the paper and finds the paper well-written, novel, insightful, and supported by a thorough ablation study.
>
> 1. ***What does "unpaired" mean in Table 2? -*** The term “unpaired” is used by the respective prior works [22, 25] to represent their weakly-supervised setting, wherein they rely on access to 2D keypoint supervision devoid of any paired 3D supervision. Please refer to Section 4.3 of HMR [22] for further details about the setup. We would like to clarify that, in Table 2, the numbers for HMR (unpaired) and SPIN (unpaired) are directly taken from the respective papers. We will update the term to “unpaired 3D” for better clarity in the revised draft.
>
> 2. ***Using a well-established regressor may play an over-important role. -*** We would like to clarify that although a regressor network trained in a supervised way seems “well-established” for a particular domain, we and several prior arts [10, 51] have reported a significant drop in performance induced by domain-shifts. This is because the supervised training biases the regressor to work only on the data domain it is trained on. For instance,
>
> * Although SPIN performs well on HR-3DPW (negligible domain-shift), we see a significant drop in performance when tested on LR-3DPW (PA-MPJPE($\downarrow$): 59.2 to 87.9). And, the proposed adaptation improves the performance to 78.9, which is a significant gain over the poor regressor performance.
>
> * Further in Table 4, Ours(S) regressor (trained on synthetic supervision) performs poorly on 3DPW, and there is a significant post-adaptation improvement PA-MPJPE($\downarrow$) 134.7 to 95.1.
>
> * The above points highlight that the synthetic supervised regressor performs significantly poorly in presence of domain shift and doesn’t play an over-important role.
>
>
> 3. ***Is inter-part occlusion an inherent limitation of the proposed method? -*** We acknowledge that inter-part occlusions are an inherent limitation of the *topology-based fitting operation* (L351). However, topology-based fitting coupled with CNN training fairly recovers this deficiency as a result of the inductive bias instilled in the source trained model (refer response #1 of reviewer fpzf for more details). Further, we are optimistic that additionally leveraging alternate modalities such as temporal or depth modalities (discern relative limb-depths) would overcome the above limitation.
>
>
> 4. ***Comment on the performance gap between the supervised and unsupervised learning for 3D human pose recovery -*** Unlike other vision tasks (such as image recognition, semantic segmentation), 3D human pose estimation is a much harder task involving apprehension of high-level semantics. The following is the prime reason behind the large performance gap.
>
> * Firstly, in 3D pose estimation the output space is usually continuous (a regression problem) as compared to the discrete output space for image recognition or segmentation (classification problems).
>
> * Secondly, beyond 2D keypoint estimation, 3D pose recovery involves a high-level semantic understanding of the articulated 3D structure (human limb articulation) and 3D view (camera position) from a single monocular image (involving inherent 2D-to-3D ambiguities).
>
> * Further, almost all the unsupervised (or self-supervised) learning for object recognition approaches [P1, P2] follow linear evaluation protocol (LEP) where labeled data is used to supervise the last classification layer to compute the final performance, which is not the case for 3D human pose estimation. We would like to distinguish the problem setting of unsupervised adaptation from the closely allied field of (completely)-unsupervised learning.
>
> * Our response #2 above, shows that available supervised models inculcate domain bias and fail to generalize in presence of domain-shift and unsupervised adaptation does provide a practical solution to this problem.
>
> 5. ***Qualitative evidence to support L332 -*** We understand that a qualitative figure to show the deficiency of OpenPose would have been more convincing. Thank you for the suggestion. Refer panel B in [Fig-link](https://drive.google.com/file/d/1bZ_onEAdVdteKvcn3o7RzR3UAcJ0fx4V/view?usp=sharing). We will include it in the revised draft.
>
> **References**
>
> [P1] Chen, Ting, Simon Kornblith, Mohammad Norouzi, and Geoffrey Hinton. A simple framework for contrastive learning of visual representations. In ICLR, 2020.
>
> [P2] He, Kaiming, Haoqi Fan, Yuxin Wu, Saining Xie, and Ross Girshick. Momentum contrast for unsupervised visual representation learning. In CVPR 2020.

---

> > ### Comment · Reviewer_xChc · 2021-08-14
> > **Post-rebuttal Comments**
> >
> > I would like to thank the authors for the fruitful discussion. My concerns are properly addressed. I thus find this paper meeting the standard of NeurIPS and it shall be accepted.
> >
> > The comments on the performance gap is especially interesting and inspiring. I agree that unsupervised learning for 3D human pose recovery has still a long way to go. Nevertheless, this journey will not be short of exciting breakthroughs.

---

> > > ### Author Response · Authors · 2021-09-02
> > > **Response to Reviewer xChc**
> > >
> > > We thank the reviewer for the encouraging feedback.

---

### Official Review · Reviewer_oYg4 · 2021-07-17

**Rating:** 6
**Confidence:** 5

**Summary:**

The paper proposes an online adaptation method to adapt pretrained models to new datasets through the silhouette and their proposed topological skeleton via inward distance map and outward distance map. They also propose a Chamfer-inspired topology-alignment objective to align the silhouette better.

**Limitations And Societal Impact:**

Will the method work on occluded cases?
like images in https://arxiv.org/abs/1905.07718 or https://prox.is.tue.mpg.de ?

Why the loss function only involves T as Eq 4?
Can D and S as loss function?

The writing and organization are poor and a little hard to follow.
For examples, citation [5] [6] is the same person, however, the names are cited differently.
line 197: double the.

**Main Review:**

The results of the paper are compelling and the generalization is good. The optimization on Thermal and low-res images is also amazing. However, as the loss function is only limited to T, I am not sure why not added the previous D and S as loss functions?

**Time Spent Reviewing:**

3

---

> ### Author Response · Authors · 2021-08-10
> **Response to Reviewer oYg4**
>
> We thank the reviewer for the valuable feedback. We are glad that the reviewer finds our results compelling and appreciates the generalization to thermal and low-res images. Below we address the reviewer’s concerns.
>
> 1. ***Will the method work on occluded cases? -*** The occluded cases mentioned by the reviewer pertain to object-induced occlusions. We would like to highlight that handling complex occlusions such as the ones caused by objects remains a significant challenge even in supervised settings.
>
> * Most of the general fully-supervised pose estimation models yield unfavorable results in the presence of occlusion without any explicit intervention. Works [P1, P2] aiming to solve this usually tackle the problem by jointly modeling human and object interactions by leveraging RGB-D data or richer external modalities.
>
> * In this work, our primary focus is to address deployment-centric domain shift issues. Although our current approach is not equipped to handle such object interactions, one can always build on this work for more challenging pose estimation scenarios (such as external object occlusion, truncation, multi-person pose estimation, etc). This remains an interesting future direction to explore.
>
> 2. ***Why does the loss function only involve $T$ (and not $D$ and $S$) in Eq 4? -*** We believe the reviewer referred to Eq 5, as Eq 4 is just a conceptual illustration of the two-way chamfer loss for the point-set representation. As mentioned in L216-222, the convolution-friendly equivalent of the same is presented in Eq 5.
>
> * We do impose separate losses (i.e., $\mathcal{L_{T}^{(sp)}}$ and $\mathcal{L_{S}^{(sp)}}$) on both $T$ and $S$ (refer last row of Table 4 and L247-250). However, we do not impose a direct loss on $D$ (in or out). Importantly, $D$ enables the realization of the alignment loss on both $T$ and $S$. Imposing the exact same loss (as in Eq. 5) directly on $D$ (in or out) is incompatible and results in spurious gradients.
>
> 3. ***Writing and organization are poor -*** We thank the reviewer for pointing out the inconsistency in a few citations, as well as the typo. We will correct these in the revised draft. As expressed by all the other reviewers, we believe the writing and organization of the paper are fairly good. We will surely put more effort into refining the organization of the paper.
>
> **References**
>
> [P1] Hassan, Mohamed, Vasileios Choutas, Dimitrios Tzionas, and Michael J. Black. Resolving 3D human pose ambiguities with 3D scene constraints. In CVPR 2019.
>
> [P2] Wang, Zhe, Liyan Chen, Shaurya Rathore, Daeyun Shin, and Charless Fowlkes. Geometric pose affordance: 3d human pose with scene constraints. arXiv preprint arXiv:1905.07718 (2019).

---

> > ### Comment · Reviewer_oYg4 · 2021-08-28
> > **Post-rebuttal Comments**
> >
> > After reading the rebuttal and other reviewers' comments I understand better about the paper. (like the generalization of the FG extraction is also a problem.). Overall I like the idea of using FG to improve the generalization ability of 3d human pose estimators.
> > My concerns are addressed!
> > and it would be better if the weakness/future work of the paper are discussed more with related work.
> > It will be nicer if authors provide failure case visualization.

---

> > > ### Author Response · Authors · 2021-09-02
> > > **Response to Reviewer oYg4**
> > >
> > > We thank the reviewer for going through our rebuttal. We will definitely include the discussion on weaknesses/future works in the revised draft.
> > >
> > > Some general failure cases are already discussed under "Qualitative results” and "Limitations” in Sec 4.2. We assure to include failure case visualizations in the presence of object-induced occlusions in the revised draft.

---

### Official Review · Reviewer_DMVp · 2021-07-17

**Rating:** 7
**Confidence:** 5

**Summary:**

The authors present a method for human pose and shape estimation that uses silhouette information as an intermediate representation that is well suited for domain shifts. Albeit not new as a concept, the proposed implementation contains several novelties (computation of distance map, efficient approximation of Chamfer-like distance metric), is sound, and effective.

**Ethical Concerns:**

n

**Ethics Review Area:**

["I don’t know"]

**Limitations And Societal Impact:**

n

**Main Review:**

Missing related work (pose estimation from silhouette, as proposed here):
Creatures great and SMAL: Recovering the shape and motion of animals from video

The approximation of distance maps via convolution and pooling operation is creative, effective, and novel. Are there any corner cases (corners?) where the approximation fails? Could you quantify the approximation to a true distance map and visualize some examples qualitatively too?

How much faster is this method to the non-convolutional computation of distance maps and chamfer distance (each measured individually). I don't doubt that it is faster, but like to know how much.

The ablation study is extensive in terms of accuracy.

Overall the paper has sufficient novelty, is easy to follow, and the evaluation is sufficient; showing clear improvements of model components and fairs well in comparison to related work.

**Time Spent Reviewing:**

1.5h

---

> ### Author Response · Authors · 2021-08-10
> **Response to Reviewer DMVp**
>
> We thank the reviewer for the constructive feedback. We appreciate that the reviewer acknowledges our paper to be easy to follow with several novelties, supported by an extensive ablation study and sufficient evaluation. Below we address the reviewer’s concerns.
>
> 1. ***Missing related work -*** We thank the reviewer for pointing out the related work [P1] which we missed. [P1] relies on silhouette as an intermediate representation while aiming to recover 3D shape of animals from videos. [P1] supervised a stacked hourglass CNN to obtain 2D joint heatmaps from binary silhouette input in order to realize the spatial-to-pointset mapping. The access to paired 2D supervision along with a fragmented training regime differentiates it from our proposed approach. We will add this under the “Use of silhouettes” subsection of related works in the revised draft.
>
> 2. ***Are there any corner cases where the approximation fails? -*** Thank you for appreciating our efforts towards realizing the distance map computation via only convolution and pooling operations. With increase in the kernel size (3$\times$3, 5$\times$5, and 7$\times$7), S2D outputs distance maps with an increase in intensity-quantization error against the true distance map (refer panel A in [Fig-link](https://drive.google.com/file/d/1bZ_onEAdVdteKvcn3o7RzR3UAcJ0fx4V/view?usp=sharing)). In our experiments, we use 3x3 kernel size as it is able to resolve and account for sharp deviations (such as at object corners or at thin foreground regions). The following table reports the quantitative comparison of the proposed approach against various true distance map definitions. The below quantitative analysis shows that our proposed approach is closer in approximation to Chebyshev (or chessboard) based distance map computation, primarily due to its convolutional nature. We will add this in the revised draft.
>
> | Comparison vs | Kernel Size | SSIM | PSNR (dB) |
> |   	-   	|              	-            	|                      	-            	|      	-                	|
> | Exact Euclidean distance map | 3$\times$3 | 0.981 ($\pm$0.08) | 31.4 ($\pm$2.7) |
> | Taxicab distance map | 3$\times$3 |0.961 ($\pm$0.21) | 24.3 ($\pm$4.1) |
> | Chebyshev distance map | 3$\times$3 | 0.996 ($\pm$0.01) | 75.8 ($\pm$1.3) |
> | Chebyshev distance map | 5$\times$5 | 0.930 ($\pm$0.07) | 17.9 ($\pm$1.9) |
> | Chebyshev distance map | 7$\times$7 | 0.882 ($\pm$0.15) | 12.3 ($\pm$2.4) |
>
> 3. ***How much faster is this method against the non-convolutional computation of distance maps? -*** Convolutional computation of the $D_\textit{in}$ takes on an average 0.1 ms, while the chamfer distance computation for T takes on an average 0.2 ms (involving computation of $D_\textit{out}$). All values are for a batch size of 16 with maps of 224x224, computed on TITAN-RTX GPU. At consistent specifications, a naive non-convolutional but parallelized implementation of the distance map takes 0.7 ms, and the Chamfer computation takes 0.8 ms.
>
> **References**
>
> [P1] Biggs, Benjamin, Thomas Roddick, Andrew Fitzgibbon, and Roberto Cipolla. Creatures great and SMAL: Recovering the shape and motion of animals from video. In ACCV 2018.

---

> > ### Comment · Reviewer_DMVp · 2021-08-25
> > **rebuttal**
> >
> > Thanks for the rebuttal, it nicely answered my questions. It would be good to include the additional insights into the paper or supplemental.

---

> > > ### Author Response · Authors · 2021-09-02
> > > **Response to Reviewer DMVp**
> > >
> > > We thank the reviewer for the positive support. We will definitely include the additional insights in the revised draft.

---

### Official Review · Reviewer_fpzf · 2021-07-23

**Rating:** 6
**Confidence:** 3

**Summary:**

Human mesh recovery methods commonly resort to 2D keypoint supervision to be able to handle in-the-wild images, as these often lack 3D ground truth. This submission argues that binary silhouettes are a better source of weak supervision and that classical segmentation techniques are less susceptible to domain shifts that 2D keypoint detectors. As such, silhouettes can be used to train shape and pose regressors on more heterogenous data without the need for annotating and retraining keypoint detectors, which this submission does: A SPIN-like method (Kolotouros et al., 2019) is proposed that replaces the keypoint-based loss and refinement, with a silhouette-based loss that allows for comparisons between the model and observed silhouettes. Key here is a differentiable skeletonisation operation that pares the silhouette down before applying the loss. The proposed method compares favourably against the state-of-the-art on a variety of datasets. The submission additionally demonstrates that the resulting scheme is more robust to adversarial perturbations than competing approaches.

**Ethics Review Area:**

["I don’t know"]

**Limitations And Societal Impact:**

The authors address the potential misuses of 3D mesh recovery for surveillance and caution against such applications.


**Main Review:**

Overall, I find this to be an interesting approach. I am somehow impressed that the submission makes the case for background subtraction as a replacement for keypoint-based fitting. In principle, I actually agree that keypoints by themselves are not reliable enough especially due to the domain adaptation problem as pointed out by the authors. The results also partly bear this out, especially when adversarial perturbations are applied. In general, incorporating low-level information can be very useful for pose estimation and indeed help address domain shifts. Silhouettes also provide information that keypoints don't include (e.g. shape, depth relations).

Nonetheless, I completely disagree with the following: [L41]: "traditional FG [foreground] segmentation techniques are mostly unaffected by input domain shifts (Fig. 1C) and thus the FG masks are usually robust and easy to obtain in diverse target scenarios". Even if we assume this to be the case, it's notoriously hard to get reliable foreground segmentations without learning. It's especially hard in crowded scenarios or in poor viewing conditions. There's also the problem of self-occlusion. This is briefly mentioned at the very of the paper in the "Limitations" paragraph, but it's a really major problem that should somehow be addressed much more explicitly when the method is being introduction. In most of the visual examples, the depicted people conveniently do not self-occlude much, but in Fig. 1 C/D, the 3rd example demonstrates the problem: The skeletonisation procedure misses the right arm entirely.

So while I agree that keypoint detection has its limitations that can partially be overcome by alternate sources of supervision, foreground segmentation (without learning) has different and partially more severe limitations. The submission also argues (or at least implies) that the lack of point-to-point correspondences in the proposed loss is an advantage over keypoint-based losses (e.g. L213), but this is not the case. If your 3D regressors provides you with a poor initialisation, the lack of correspondences is a problem. It is not a problem and can even be helpful when you are close to the solution and do not want to fit your model to observation noise but there is a trade-off.

The results are good enough for me, but I'm not sure about the fairness of the comparison against the state-of-the-art. Table 2 compares against other methods on H36M, and this approach outperforms others in the weakly- and unsupervised setting. However, it is very important especially in this area to have some comparisons that control for differences in the training data. Having the right pose distribution is at least as important as being able to handle shifts in appearance. As such, it would be good to have direct comparisons against HMR or SPIN using the same training setup. Given that SPIN is used for the ablation study later, that would probably be a good choice. The comparison against 3DPW is similarly problematic. Why aren't HMR and SPIN also compared in the weakly-supervised setting? It is not very surprising that SMPLify is significantly worse, given that it relies on optimising the method. Otherwise I think the rest of the experiments are nice, e.g. the ablation study comparing different losses as well as the ones with adversarial perturbations / challenging low-resolution images. What I'm missing however is a description of how the silhouettes are obtained on the UAP-H36M and LR-3DPW datasets. This is a critical detail that will help me judge these results better.

Silhouettes have a very long history in this area, and it's great that the submission cites some older papers. However, there are a lot of other relevant references that need to be taken into consideration. These need not all be cited, but I mention them here because of certain inaccuracies in the text: One very early method that is relevant is the 2002 paper of Sminchisescu and Telea "Human Pose Estimation from Silhouettes: A Consistent Approach Using Distance Level Sets". If I'm not mistaken, they also use something very similar to the "Distance-map" described in L153-L159.  Figure 3 in that paper shows the "outwards distance-map" as it is referred to in this submission, and the full objective is described in section 4. As such, it is inaccurate to that this has been "introduce[d]" here (L152). I also think the description of "chamfer[-]based distance[s]" (L145-L149) is inaccurate, as many rely on distance transforms as well, e.g. Gavrila & Davis (1996) "3-D Model-based tracking of humans in action: a multi-view approach" (Fig. 6 and section 4.1), described in more detail I think in Gavrila (1998) "Multi-feature Hierarchical Template Matching Using Distance Transforms": "The chamfer distance consists thus of a correlation between T and the distance image of I, followed by a division." A couple of newer methods that use silhouettes include Alldieck et al., (2017) "Optical Flow-based 3D Human Motion Estimation from Monocular Video" who use a distance-transform-based loss as well, and Omran et al. (2018) who use part segmentations as an intermediate step before lifting to 3D.

Just to be clear, I understand that a key difference here is the skeletonisation step, rather than the direct comparison of distance transforms applied to silhouettes, and that this extra step is helpful here (Table 4). The text however still needs to be amended as suggested. It would also probably not hurt to refer to a few works on skeletonisation based on distance transforms, as this also has a long history, e.g. Wright et al. (1995) "Skeletonization using an extended Euclidean distance transform" or Latecki et al. (2007) "Skeletonization using SSM of the Distance Transform". Maybe there are more appropriate references. What appears to be novel here as well is the differentiable skeletonisation but I am not sufficiently familiar with this problem area not to mention the use of distance transforms and skeletonisation outside of pose estimation, so I'm not sure.

I'm not sure why the binary activation function described in section 3.2 is necessary. The submission justifies this by stating that "a non-binary silhouette is not suitable for distance-map and topology definitions" (L176). Why would a "silhouette obtained via a fast differentiable renderer" (L174) be non-binary? If every mesh vertex has the same colour and no lighting is enabled, the resulting image will be binary and would not break differentiability.

Editing suggestions:
- the caption in Figure 2: The description of the topological-skeleton extraction in B is also in 3.1.3 (not just 3.2).


**Time Spent Reviewing:**

3

---

> ### Author Response · Authors · 2021-08-10
> **Response to Reviewer fpzf**
>
> We thank the reviewer for the detailed and constructive feedback. We appreciate that the reviewer identifies our work as an interesting approach and acknowledges our key novelty of realizing differentiable extraction of topological skeletons. Below we address the reviewer’s concerns.
>
> 1. ***2D-detection has issues, however, FG segmentation (w/o learning) has partially more severe limitations -*** We agree with the reviewer that FG segmentation techniques (w/o CNN-based learning) would severely fail (L349) for the following scenarios: crowded scene, poor viewing conditions, self-occlusion, etc. As briefly mentioned in L351, though an only fitting-based approach severely fails to solve such cases, optimization coupled with CNN training fairly recovers this deficiency as a result of the inductive bias instilled in the source trained model.
>
> * ***Leaning in presence of hard scenarios -*** Though a given target dataset may contain such hard scenarios (noisy FG segmentation label), the approach only requires 60-70% of samples with clean FG segmentation label in order to adapt the source trained model to the target environment. Gradient descent fits to clean labels significantly faster than the noisy ones, refer Fig. 1a of [P1]. Another way to address this would be to enhance the source inductive bias by training the source model on synthetically simulated hard scenarios (labeled). Note that, the adaptation training only updates the environment-specific factors while holding on to the source inductive bias.
>
> * Further, the quantitative comparison in Table 5 (A1 or A2 vs Ours) shows that FG predictions obtained from traditional FG segmentation techniques (refer response #2 below) are more reliable than the 2D keypoint predictions obtained from the 2D-detector CNNs for an effective self-adaptation performance.
>
> 2. ***How are the silhouettes obtained on UAP-H36M and LR-3DPW? -*** The silhouette predictions are obtained using a traditional BG subtraction technique, MOG-BG [P2] integrated with results from Grab-Cut [P3]. This technique works well under static camera scenarios, as in H36M. However, for 3DPW, we discard the frames with severe camera movements.
> We understand that this is an important detail, which we missed. We will update it in the revised draft.
>
> 3. ***The submission argues that the lack of point-to-point correspondence is an advantage over keypoint-based losses -*** We would like to clarify that, in L213, we do not mean lack of correspondence (and the unmatched set cardinality) as an advantage. Rather, we meant to highlight it as a disadvantage (or a hurdle) that needs to be addressed to effectively realize the proposed alignment loss. We will explicitly clarify it in the revised draft.
>
> 4. ***Why aren’t HMR and SPIN compared in 3DPW weakly-supervised settings? -*** Though we directly took the numbers for HMR and SPIN from the respective papers for H36M (weakly sup. in Table 2), the same numbers (or trained weights) for 3DPW (weakly sup. in Table 3) have not been provided by the respective papers [22, 25].
>
> 5. ***Comparisons that control for the differences in training data -*** We understand the reviewer’s concern. However, the weakly sup. training regime is not provided in the official releases (of SPIN and HMR) and implementing the same from scratch would have several inconsistencies. We will try our best to include it in the revised draft. Nonetheless, the paper’s primary focus is to evaluate our self-adaptive approach in the presence of drastic domain shifts, which is well analyzed in Table 5.
>
> * We would like to highlight that most prior-art methods report such comparisons despite having differences in training data. This is generally accepted in the community as long as the advantages and disadvantages are clearly reported. For instance, **a)** In Table 1 and 3 of SPIN paper [25], HMR(unpaired) and SPIN(unpaired) are compared in the same table. However, SPIN(unpaired) is clearly at a disadvantage against HMR(unpaired) as listed in the table below. **b)** Similarly, in Table 1 of Sim2real [10], Sim2real compares against HMR and Martinez et al. [34] despite having significant training data differences (such as a custom in-house dataset). The following table lists the dataset differences:
>
> | Weakly sup. methods | Datasets used (paired) | Datasets used for Pose-prior |
> |   	-   	|              	-            	|               	-                	|
> | HMR (unpaired) [22] | H36M , MPI-3DHP, LSP, LSP-extended, MPII, and COCO | CMU-MoCap, Pose-Prior, H36 training-set |
> | SPIN (unpaired) [25] | H36M, MPI-3DHP, LSP, LSP-extended, MPII, and COCO | CMU-MoCap |
> | Doersch et al. [10] | COCO, SURREAL, Custom in-house data | - |
> | Ours(S→R) | SURREAL, H36M, MPII | CMU-MoCap |
>
> 6. ***Inaccurate use of “introduce” in L152 -*** Owing to the vast amount of literature pertaining to distance-transforms, there exist several conflicting formulations and definitions. Hence, in L152 (and the subsequent subsections), we use the word “introduce” to convey the relevant definitions to the reader for a better understanding of the proposed approach. We will rephrase it to better reflect our intent.
>
> 7. ***Inaccurate description of chamfer in L145 -***  We thank the reviewer for suggesting older works relating to the use of silhouettes and distance transforms. In L145, the description of Chamfer is mentioned in the context of “silhouette supervision for human mesh recovery”. We understand it might confuse the reader and will amend the statements in the revised draft while citing the older works. We will add these suggestions in Sec 2 under “use of silhouettes”. To the best of our knowledge, none of the prior works facilitate differentiable convolution-friendly computation of topological skeletons for self-adaptation.
>
> 8. ***Why would a silhouette obtained via a fast differentiable renderer be non-binary? -*** Differentiable renderers which aim to approximate the rasterization process in a soft and differentiable way tend to yield non-binary pixels at the edges of the rendered object. This is despite each mesh vertex being assigned a fixed pixel intensity with lighting disabled (refer Sec 3 and Fig 2 of PyTorch3D [42]).
>
> We will address all other minor concerns and suggestions in the revised draft.
>
> **References**
>
> [P1] Chiyuan Zhang, Samy Bengio, Moritz Hardt, Benjamin Recht, and Oriol Vinyals. Understanding deep learning requires rethinking generalization. In ICLR 2017.
>
> [P2] Pakorn KadewTraKuPong and Richard Bowden. An improved adaptive background mixture model for real-time tracking with shadow detection. In AVBS 2001.
>
> [P3] Carsten Rother, Vladimir Kolmogorov, and Andrew Blake. GrabCut - Interactive Foreground Extraction using Iterated Graph Cuts. In SIGGRAPH 2004.

---

> > ### Comment · Reviewer_fpzf · 2021-08-28
> > **Clarification re: 3DPW evaluation**
> >
> > Thanks for the detailed response.
> >
> > [Q2] " This technique works well under static camera scenarios, as in H36M. However, for 3DPW, we discard the frames with severe camera movements. [...] We understand that this is an important detail, which we missed. We will update it in the revised draft."
> >
> > Could you provide more details here? How many frames were discarded? I don't understand how this went unmentioned. Did you exclude the same frames for other methods in Table 3? There are many sequences in 3DPW where the camera is moving, especially in the downtown scenarios - which are also rather crowded. It's hard for me to see how a GrabCut-based background subtraction approach can work well enough in these settings. I'm not opposed to modifying the evaluation procedure in principle, but any such modificatio needs to be described in more detail than this.

---

> > > ### Author Response · Authors · 2021-08-28
> > > **Clarification on 3DPW evaluation**
> > >
> > > Thank you for going through our rebuttal. Below we clarify the concerns.
> > >
> > > > How many frames were discarded? I don't understand how this went unmentioned. Did you exclude the same frames for other methods in Table 3?
> > >
> > > **1.** We would like to clarify that our evaluation procedure (inference on test-set) is exactly inline with the prior established standards (as in [25]). Thus, our evaluation considers the same frames as used by other methods in Table 3.
> > > * `Note that in Table 3, none of the approaches (including ours) use 3DPW for training (refer to Table 3 caption).`
> > > * `We discard frames only for the LR-3DPW training-set, i.e., for adaptation of R to LR-3DPW in Table 5. And, this remains consistent across all the baselines reported in Table 5.`
> > >
> > > > There are many sequences in 3DPW where the camera is moving, especially in the downtown scenarios - which are also rather crowded. It's hard for me to see how a GrabCut-based background subtraction approach can work well enough in these settings.
> > >
> > > **2.** To the best of our knowledge, all the prior single-person 3D human pose estimation approaches [22, 25, 51] (including the unsupervised [10, 26] and weakly supervised approaches [4, 10, 25]) assume access to detector output to crop the subject of interest. And, we also follow the same. Thus, even in crowded scenarios, only the subject of interest is considered to obtain reliable background subtraction.
> > >
> > > > I'm not opposed to modifying the evaluation procedure in principle, but any such modification needs to be described in more detail than this.
> > >
> > > **3.** We would like to clarify that our evaluation procedure (inference on test-set) is exactly inline with the prior established standards (as in [25]). Note that background subtraction is used only for the training set samples and does not come into picture during evaluation.

---

> > > > ### Comment · Reviewer_fpzf · 2021-09-02
> > > > **3DPW evaluation**
> > > >
> > > > I'm unable to reconcile the following two statements from the discussion:
> > > >
> > > > "1. We would like to clarify that our evaluation procedure (inference on test-set) is exactly inline with the prior established standards (as in [25]). Thus, our evaluation considers the same frames as used by other methods in Table 3."
> > > >
> > > > "However, for 3DPW, we discard the frames with severe camera movements."
> > > >
> > > > Which is it?

---

> > > > > ### Author Response · Authors · 2021-09-02
> > > > > **Clarification on 3DPW evaluation**
> > > > >
> > > > > Background subtraction (and discarding unsuitable frames for background subtraction) is used only for the training set samples. In Table 3, none of the approaches use 3DPW for training.
> > > > >
> > > > > Note that 3DPW is used as a training set only for experiments in Table 5 (i.e., adaptation of R to LR-3DPW). Thus, the second statement is applicable for experiments in Table 5 and not in Table 3.
> > > > >
> > > > > Also, refer to the bullet points under our response #1 above. We hope this clarifies your concerns.

---

### Decision · Program_Chairs · 2021-09-27

**Decision:**

Accept (Poster)

**Comment:**

This submission has received 4 positive final ratings: 6, 7, 6, 7.
The reviewers appreciated overall novelty, extensive experiments, strong empirical performance and clear presentation.
The remaining minor questions and concerns were addressed by the authors in the rebuttal, as acknowledged by the reviewers.
As a result, the final recommendation is to accept for poster presentation.